



# Investigation of Aerosol Transport Flux Structure over Beijing Based on Lidar Observations and the Impact of Dust Transport on Air Quality

Zhengguo Tian[1], Longlong Wang[1,*], Si Liu[1], Cheng Yao[1], Tong Lu[2], Weijie Zou[2], Zhenping Yin[2], Yun He[3,6], Lei Li[4], Bin Zhang[1], Daru Chen[1], Zhichao Bu[5], Yubao Chen[5], Xuan Wang[2]

[1]Hangzhou Institute of Advanced Studies, Zhejiang Normal University, Hangzhou, 311231, China
[2]School of Remote Sensing and Information Engineering, Wuhan University, Wuhan, 430079, China
[3]School of Earth and Space Science and Technology, Wuhan University, Wuhan, 430072, China
[4] State Key Laboratory of Severe Weather Meteorological Science and Technology & Key Laboratory of Atmospheric Chemistry of CMA, Chinese Academy of Meteorological Sciences, Beijing 100081, China
[5]Meteorological Observation Center, China Meteorological Administration, Beijing, 100081, China
[6]State Observatory for Atmospheric Remote Sensing, Wuhan 430072, China.

*Correspondence to*: Longlong Wang (longlong.wang@zjnu.edu.cn)

**Abstract.** The origins, spatial distribution, and diffusion mechanisms of aerosols hold practical guiding significance for regional haze governance. The vertical and horizontal fluxes of aerosols serve as effective parameters for assessing the diffusion efficiency of aerosols, but they are less exploited due to insufficient observations. This study uses polarization lidar to differentiate between the aerosol sources of dust and non-dust and to estimate the mass concentration profiles of each. Combining the wind profiles acquired from Doppler wind lidar, the vertical and horizontal fluxes profiles of two type aerosols are calculated. This approach is designed to account for the influence of local aerosol transport mechanisms on air pollution, enabling a more precise reflection of the internal variations within a particular region. A winter haze event in Beijing from November 2nd to November 4th, 2023 was analyzed to distinguish the vertical distribution of and mass concentration and fluxes of aerosols brought by dust transported from the northwest monsoon and those from anthropogenic emissions within the North China plain area. Further analysis of different wind zones revealed that the aerosol concentration and fluxes from urban areas (regions with a higher density of anthropogenic sources) can be used to assess the local pollutant diffusion capacity, as well as the influence of vertical turbulence on ground $PM_{10}$ concentrations. Taking Beijing as an example, this study investigated the diffusion characteristics of urban aerosols, ultimately providing technical means and data references for early warning of urban air pollution and assessment of air pollution control measures.

## 1 Introduction

Atmospheric particulate matter or aerosols are fine solid or liquid particles that come from direct emissions or secondary conversion of gas - particulate matter and can suspend in the air. Atmospheric particulate matter pollution is one of the issues that the whole world pays attention to, and there is sufficient evidence showing that it has adverse impacts on human daily life



and health (Brunekreef et al., 2002). Anthropogenic aerosol emissions are one of the primary sources of air pollution. Klimont et al. (2017) estimated that the global anthropogenic total emissions of $PM_{2.5}$ in 2010 were approximately 80 million tons, among which anthropogenic combustion aerosols (including transportation and industrial emissions) accounted for more than

80% (Klimont et al., 2017). China, with its vast territory, high population density and rapid economic development, has inevitably produced a large amount of anthropogenic aerosols during its development process, which has had a significant impact on air quality in East Asia. Since 2013, China has dedicated substantial efforts to air pollution governance and has promulgated numerous measures for managing and controlling aerosol emissions in cities (The State Council of the People's Republic of China, 2013; Geng et al., 2019). Nevertheless, these measures are primarily centered on the overall supervision of

emission volumes. There exists an absence of a holistic approach to address various aspects of aerosol emissions, including the temporal dynamics, spatial patterns, dispersion processes, and the principal determinants that significantly contribute to air pollution. As a result, it remains challenging to establish an accurate and timely early warning system. Consequently, research on the emission and diffusion mechanisms of urban anthropogenic aerosols holds significant practical guiding value for the effective formulation and evaluation of emission policies.

45        Aerosols can be lifted from the Earth's surface into the atmosphere due to factors such as deep convection, frontal passages, and turbulence within the atmospheric boundary layer (Chen et al., 2012). This vertical transport mechanism is crucial for the initial step of regional-to-intercontinental propagation. It is also of great significance for gaining an in-depth understanding of the frequent occurrence of long-duration and large-scale air pollution in China, as well as for improving urban air quality across the country. For example, it has revealed a new mechanism by which severe haze pollution in the Beijing-Tianjin-Hebei

and Yangtze River Delta urban agglomerations during autumn and winter is intensified through mutual interaction and long-distance transport (Huang et al., 2020). Meanwhile, this can provide new insights for mitigating regional-scale mutual influences, thereby achieving scientific and precise haze control.

       Atmospheric pollution in urban areas is mainly caused by anthropogenic emissions, including vehicle exhaust emissions, industrial emissions, and the combustion of carbon substances in daily life. Therefore, anthropogenic aerosols in urban areas

can serve as an "emission indicator" for air pollution sources, which can be used to evaluate the effectiveness of air pollution reduction and air quality management measures. In the causes of air pollution, in addition to pollutant emission sources, the atmospheric planetary boundary layer height (PBLH) also plays an important role in controlling the temporal and spatial variations of pollutant concentrations (Ding et al., 2015). In most cases, anthropogenic air pollutants emitted from the surface are confined within the atmospheric planetary boundary layer, and their vertical mixing height directly determines the vertical

mixing space of pollutants (Lou et al., 2019; Gerbig et al., 2008).

       The interaction between vertical atmospheric pollution and the atmospheric boundary layer feedback process is an important factor in enhancing the transport of pollution between regions. The vertical flux of atmospheric aerosols is a key physical quantity used to quantitatively characterize the net transport of aerosols in the vertical direction per unit area and per unit time. It connects surface processes, boundary layer turbulence, aerosol cycling, and meteorological conditions. Its core

significance lies in quantifying the intensity and direction of aerosol transport in the vertical direction, thereby revealing the



source and sink mechanisms of aerosols, their diffusion capabilities, and regional environmental impacts (Huang et al., 2024). Therefore, accurately assessing aerosol fluxes depends on the characteristics of the aerosol sources, the regional features, and the meteorological conditions within the atmospheric boundary layer. As Huang et al. (2020) demonstrated, aerosols themselves, especially light-absorbing components such as black carbon, can significantly influence the boundary layer

structure through Aerosol-Boundary Layer Interaction (ABI), and amplify the accumulation and transport processes of regional pollutants (Huang et al., 2020). Due to the complexity and nonlinearity of the causes of atmospheric compound pollution under China's unique environmental conditions (Lv et al., 2017), more stringent requirements are placed on detection tools and methods in aerosol flux research. Typically, a comprehensive assessment of aerosol flux necessitates the employment of a combination of various methodologies and technologies, encompassing ground-based observations and vertical stereo-

detection techniques. Moreover, it is difficult to estimate the emission of particulate matter and the capacity of dry deposition at the surface to remove particulate matter from the air, which makes it impossible to parameterize these processes for application in atmospheric numerical simulation techniques.

Horizontal transport caused by wind speed and wind direction is also an important factor restricting the mixing space of pollutants (Weissmann et al., 2005; Wang et al., 2009; Tian et al., 2020; Zhang et al., 2020; Hou et al., 2022; Lin et al., 2023).

However, in the actual assessment and prediction of air pollution, it is difficult to consider the three factors of emission volume, atmospheric boundary layer, and dominant wind speed in a coordinated manner. Consequently, the horizontal flux can offer crucial insights into the horizontal transport of aerosol sources within cities and the net emissions resulting from external sources. For example, the lidar observations in Singapore (Huang et al., 2024) have shown that cross-border haze originating from biomass burning in Indonesia is transported via the nocturnal low-level jet. During the development of the daytime

boundary layer, it is entrained to the ground surface, leading to a 500% increase in the local $PM_{2.5}$ concentration (Huang et al., 2024). Conversely, the findings of Zhang et al. (2015) suggest that the prevailing wind exerts a substantial influence on the pollution concentration in cities. For instance, the northerly wind is correlated with relatively lower pollutant concentrations in Beijing (Zhang et al., 2015).

Over the past two decades, many scholars have attempted to use different observation methods to conduct targeted studies

on the sources, diffusion mechanisms, spatial distribution, and other important influencing factors of particulate matter pollution (Tanaka et al., 2005; Weissmann et al., 2005; van Donkelaar et al., 2016; Huang et al., 2024). In the field campaigns of aerosol observation techniques, in-situ continuous monitoring remains one of the principal detection approaches. Its advantage is the ability to provide a high temporal resolution and straightforward aerosol mass concentration data. In-situ sampling monitoring can not only offer a relatively comprehensive understanding of aerosol characteristics but also be utilized

to analyze the composition of particulate matter. Additionally, a network of such sampling points can be established for research on regional near-surface particulate matter emissions (Liu & Daum, 2000; Drinovec et al. 2015). Currently, numerous methods for measuring the surface-atmosphere exchange processes of aerosols have been developed (Pryor et al., 2008). Among these, the most representative, direct, and productive systematic observational instruments and data analysis techniques for aerosol fluxes are associated with the European ACTRIS (Aerosol, Cloud and Trace Gases Research Infrastructure)



program. This program utilizes standardized field measurement instruments installed on meteorological towers less than 100 meters in height. For example, by combining ultrasonic anemometers and particle size spectrometers to obtain single-point aerosol fluxes. However, this approach can only represent relatively small areas. As a result, when applying the aerosol flux results to mesoscale circulation models, significant uncertainties may occur (Chen et al., 2008). Consequently, there is an urgent demand for vertical profile observations of aerosol fluxes that span the entire atmospheric boundary layer.

In recent years, remote sensing technology, particularly lidar, has become a powerful tool for aerosol research due to its high spatiotemporal resolution and detection sensitivity (Li et al., 2019; Lopatin et al., 2013; Che et al., 2019; Wang et al., 2019; Weinzierl et al., 2011; Denjean et al., 2016; Córdoba-Jabonero et al., 2018). Unlike passive sensors, lidar systems actively probe the atmosphere and can retrieve vertically resolved aerosol properties—such as depolarization ratio, extinction coefficient, and mass concentration—along with wind field information within the planetary boundary layer (PBL) (Müller et

al., 2001; Böckmann et al., 2001; Mamouri & Ansmann, 2014). The combination of multi-wavelength polarization Raman lidar and Doppler lidar offers a particularly advanced approach for quantifying aerosol fluxes. By leveraging covariance algorithms and co-located measurements of aerosol properties and 3D wind vectors, this synergistic technique enables direct estimation of both vertical and horizontal aerosol transport with high accuracy (Engelmann et al., 2008; Huang et al., 2024). Thus, the integration of these lidar technologies provides a feasible and robust method for studying aerosol fluxes within the

atmospheric boundary layer. However, most of the current detection methods and contents have the following deficiencies: 1. Insufficient detection methods and research on anthropogenic aerosol emissions. Most of the current flux research methods use in-situ or passive remote sensing observations (e.g. Yuan et al., 2019; Conte & Held, 2021; Casquero-Vera et al., 2022; Hu et al., 2022), lacking high-resolution vertical spatial scale variation results, and rarely estimating natural and anthropogenic aerosols (dust and anthropogenic emissions) separately; Insufficient consideration of regional meteorological conditions on

the concentration of atmospheric pollutants, it is necessary to consider the impact of both horizontal and vertical mixing on air pollution; Currently, there is a lack of effective quantitative assessment of the sources and sinks (accumulation and removal rates) of atmospheric pollutants. Most importantly, urban aerosol fluxes largely depend on site characteristics, so more urban particulate matter flux data from different cities are needed to fully understand the role of urban pollution sources and their spatial variability.

This paper focuses on an air pollution event in Beijing during winter and conducts a detailed analysis of the impact of wind direction shear on aerosol transport within a short period of time. Polarization lidar and Doppler wind profile lidar were used to obtain the vertical and horizontal aerosol mass concentration flux profiles to consider the influence of local aerosol transport mechanisms on air pollution. The observation results can be used to accurately reflect the internal changes in a specific area, and ultimately provide technical means and data references for early warning of urban air pollution and

assessment of air pollution control measures, such as in large cities like Beijing. The northwest of Beijing is mountainous, and the southeast is a plain. The plain slopes gently from northwest to southeast, with the height gradually decreasing. Due to the influence of topography and climate, a large amount of air pollutants is not only emitted in the urban area of Beijing, but also in the suburbs, rural areas, and surrounding cities. The mutual exchange of air pollutants among these cities makes the





understanding of the formation of air pollution and the formulation of control strategies more complex. As a hot spot
representative of the central area of the Beijing-Tianjin-Hebei region, the emission characteristics and meteorological
conditions of the urban area of Beijing are different from those of other urban agglomerations. Therefore, the aerosol flux may
differ from the observation results of other urban agglomerations.

## 2 Instrumentation and methodology

### 2.1 Doppler wind lidar

Wind lidar is the most effective method for global wind field information detection and can also obtain wind field
information within the three-dimensional profile. In this study, a Doppler wind lidar system was used to monitor the vertical
profile of the wind field. The detection range of this system is 3000 m, with a distance resolution of 50 m and a time resolution
of 1 minute. During the long-term observation period, the lidar system was set to the velocity azimuth display (VAD) scanning
mode, with a fixed elevation angle of 60°, an azimuth range of 0° to 300°, a scanning interval of 5°, and a total of 60 radial
profiles were obtained for each scanning circle, lasting for 135 seconds. Then, based on the assumption of a horizontally
uniform wind field, the horizontal wind speed, wind direction, and vertical wind speed were extracted from the measured radial
velocities at different azimuth angles (Smalikho, 2003; Banakh et al., 2013).

### 2.2 Raman–Mie scattering aerosol LiDAR (REAL)

REAL is a single-wavelength polarized Raman-Mie scattering aerosol lidar with a wavelength of 532 nm, featuring a
groundbreaking design that achieves a near-surface blind zone of less than 50 meters—significantly below the industry-typical
blind zone of over 100 meters—which provides a crucial advantage for high-precision joint analysis with ground-based
observational data. The system's receiver is equipped with six channels, among which the core channels include the 532 nm
parallel polarization channel (P), the 532 nm perpendicular polarization channel (S), and the 607 nm nitrogen Raman channel.
This configuration, as detailed by Chen et al. (2024) regarding system parameters, enables the REAL to possess continuous
observation capabilities from the ground to the upper atmosphere. To ensure the dynamic range of the data from the three main
receiving channels, each main channel is divided into high and low sub-channels through a non-polarizing beam splitter and
data gluing is achieved through linear overlap (Zhang et al., 2014). To enhance the dynamic range of the lidar echo signal, in
the system design, optimization was made in terms of stray light, anti-electromagnetic interference in the detection circuit, and
polarization isolation. After the design was completed, it passed a series of tests of mutual comparison with the 532 nm
reference lidar MUSA (Wandinger U, et al., 2016) of the EARLINET lidar calibration center with high quality. In the mutual
comparison, it was found that the signal deviation of REAL and MUSA within the detection dynamic range was less than 2%,
and the dynamic detection range was larger.

In this research, the $PM_{2.5}$ and $PM_{10}$ data employed were primarily sourced from the official website of the China National
Environmental Monitoring Centre (http://www.cnemc.cn/). This website offers real-time dissemination and historical data



query capabilities for air quality monitoring data across different regions in China. Furthermore, air quality-related information (Beijing's PM concentration) published on the official website of the Beijing Municipal Ecology and Environment Bureau (http://sthj.beijing.gov.cn/) was also consulted for supplementation and verification purposes. When analyzing the direction and backward trajectories of long-range transported aerosols, we utilized the Hybrid Single-Particle Lagrangian Integrated Trajectory (HYSPLIT) model (https://ready.arl.noaa.gov/HYSPLIT.php). The HYSPLIT model has been extensively applied

in calculating the trajectories of air quality, pollutants, and their transport processes (Kim et al., 2020).

### 2.3 Data processing

In this study, the aerosol optical properties were retrieved using a 532-nm single-wavelength Raman-Mie polarization scattering lidar (REAL) system, and the wind profiles were obtained by means of a Doppler wind lidar. The raw dataset measured by the REAL is sampled within 1 minute, with a vertical resolution of 15 m. Before retrieving aerosol optical

properties, these data undergo preprocessing steps, including dead time correction, background noise subtraction, signal gluing and averaging, range correction, and polarization gain ratio calibration to ensure the quality of the data (Freudenthaler, 2016; Wang et al., 2023; Mao et al., 2024). The aerosol backscattering coefficient is retrieved using the sum of the horizontally and vertically polarized Mie scattering signals (D'Amico et al., 2015).

Since the presence of clouds can severely bias the retrieval of aerosol optical properties, cloud-affected range bins were

identified and masked before inversion using the Value Distribution Equalization (VDE) algorithm originally proposed for ground-based lidar (Zhao et al., 2014). In VDE, semi-discretization is applied to suppress random noise while preserving sharp layer gradients, followed by histogram-equalization to enhance far-range signal visibility; empirical gradient/attenuation tests are then used to delineate cloud layers. The practical reliability of this screening under diverse atmospheric conditions has been demonstrated in long-term, multi-instrument evaluations (Zou et al., 2024). Only cloud-free bins were used for the

subsequent aerosol inversions and particle depolarization ratio (PDR)-based classification.

The classification of dust and non-dust aerosols is realized based on the PDR (Freudenthaler et al., 2009).

$$\delta_p = \frac{(1+\delta_m)(\delta_v+1)\beta_a}{\beta_m(\delta_m-\delta_p)+(1+\delta_m)\beta_a} - 1 \ , \tag{1}$$

In the text, $\delta_m$ represents the depolarization ratio of the molecules, $\delta_v$ denotes the volume depolarization ratio, and $\beta_a$ and $\beta_m$ are the backscattering coefficients of aerosols and molecules, respectively. Subsequently, classification thresholds are set:

when $\delta_p \geq 0.31$, it is identified as pure dust aerosol ($\delta_{dust}$); when $\delta_p \leq 0.05$, it is identified as non-dust aerosol ($\delta_{non-dust}$); the intermediate range is classified as a mixture of dust and other aerosols. The backscattering coefficient of dust is calculated following these rules: In a pure dust region, the total backscattering coefficient is solely contributed by dust. In a non-dust region, the contribution of dust is zero. In a mixed region, based on the linear mixing model, the contribution of dust is proportionally distributed according to the PDR. (Tesche et al., 2009; He et al., 2023):

$$\beta_{dust} = \beta_a \cdot \frac{\delta_p - \delta_{non-dust}}{\delta_{dust} - \delta_{non-dust}} \cdot \frac{1+\delta_{dust}}{1+\delta_p} \ , \tag{2}$$





The total backscattering coefficient minus the backscattering coefficient of dust yields the non-dust backscattering coefficient.

The aerosol mass concentration is derived via the joint calculation of the extinction coefficient and the mass concentration conversion factor (Mamouri & Ansmann, 2014).

$$m(z) = \beta_{non-dust/dust}(z) \cdot LR \cdot c \,, \tag{3}$$

Among them, the aerosol extinction coefficient which is calculated by dust and non-dust backscattering coefficient with assumed fixed lidar ratio ($LR$), and $c$ represents the mass concentration conversion factor. In this study, the mass concentration conversion factor $c$ for dust aerosols is adopted as $2.002 \times 10^{-3}$ kg·m$^{-2}$, while that for non-dust aerosols is taken as $0.6355 \times 10^{-3}$ kg·m$^{-2}$ (Mamouri & Ansmann, 2017).

The vertical aerosol mass flux is calculated using the eddy covariance method (Swinbank, 1951). The eddy covariance method is a direct measurement method for vertical fluxes based on turbulence theory. Thus, the aerosol mass concentration vertical fluxes at different heights can be expressed as follows (Engelmann et al., 2008; Dipu et al., 2013):

$$F_m = \overline{m'(z)v'(z)} \,, \tag{4}$$

When calculating the vertical flux, the instantaneous fluctuation of the vertical wind speed at a given altitude is defined
such that upward is positive and downward is negative. A positive vertical flux implies that the vertical motion is responsible for aerosol transport, while a negative vertical flux indicates that the vertical motion is involved in aerosol scavenging. While the aerosol mass concentration horizontal fluxes at different heights can be expressed as follows (Hu et al., 2022; Lin et al., 2023; Li et al., 2024):

$$F_h = \overline{m(z)} \times \overline{w(z)} cos(wd) + \overline{m'(z)w'(z)} \,, \tag{5}$$

Among them, $z$ represents the vertical range of the lidar. The time-averaged value (30 minutes) is calculated by computing the covariance between the specific aerosol mass concentration $m'(z)$ and the fluctuation $v'(z)$ of the vertical wind speed at different altitudes caused by turbulence; the time-averaged horizontal wind speed $\overline{w(z)}$ and aerosol concentration $\overline{m(z)}$ (30 minutes) are used for calculation. In addition, the covariance between the specific aerosol mass concentration $m'(z)$ and the fluctuation $m'(z)$ of the horizontal wind speed at different altitudes caused by turbulence is also considered in the horizontal
fluxes in this study. When computing the horizontal flux, we decompose the horizontal wind speed into components in the north-south direction $cos(wd)$, where the northwestward direction is defined as positive. When there is a positive correlation between the wind direction and the fluctuations in aerosol mass concentration, the horizontal flux is positive, signifying the northward transport of aerosols. The column of both fluxes is calculated as the product of the transport flux per unit cross-sectional area and the sum of the heights of each layer (Hu et al., 2022):

$$F_c = \sum F_m \times z_i \,, \tag{6}$$



Among them, $z_i$ represents the height interval of each layer. This study further calculates the emission rates of dust and non-dust aerosols, which are derived by dividing the aerosol flux by their respective mass concentrations, thereby directly reflecting the transport efficiency per unit concentration of aerosols.

## 3 Result Analysis

The observation site is located at the Beijing South Suburban Observatory (latitude: 39°54'20" N, longitude: 116°25'29" E). The observation period is selected from 19:30 China Standard Time (CST) on November 2, 2023 to 9:30 CST on November 4, 2023, with an interval of 38 hours. Throughout the observation, the sky was predominantly clear or had sparse cloud cover. Only the contributions of optical properties caused by aerosols are used so that ensured the continuous acquisition of aerosol vertical profiles and wind field information over the free troposphere. The concentration of ground-level particulate matter

exhibits significant temporal variations: from the evening to the night of November 2nd, the $PM_{2.5}$ and $PM_{10}$ concentrations reached a high peak value, with the $PM_{2.5}/PM_{10}$ ratio exceeding 0.6, indicating that the aerosol was predominantly composed of fine particles. Starting from November 3rd, both $PM_{2.5}$ and $PM_{10}$ concentrations showed a declining trend, and the ratio also decreased sharply Figure 1(a).

At 00:00 CST on November 2nd, the near-surface $PM_{10}$ concentration in Beijing reached 200 μg/m³, with a $PM_{2.5}/PM_{10}$

ratio of 0.8, indicating that the pollution event was primarily dominated by local emissions. The diurnal variation in boundary layer height and the modulating effect of wind shear on the distribution of the aerosol layer can be identified in the figure 1(b-f), providing intuitive observational evidence for further understanding the vertical diffusion and transport mechanisms of regional pollutants. As depicted in Figure 1(b), the RCS532 profile simultaneously revealed a relatively pronounced high aerosol concentration within the lower layer (0-1.5 km) at 19:30 CST on November 2nd. Concurrently, the particle

depolarization ratio ($\delta_p$) shown in Figure 1(b) increased to 0.3-0.4, suggesting that the aerosol particles exhibited strong non-spherical characteristics. The combination of a high backscattering coefficient and elevated depolarization ratio observed during this period closely matches the optical fingerprint of typical dust aerosols (Peng et al., 2021; Floutsi et al., 2023; Tesche et al., 2009). Furthermore, the $PM_{2.5}/PM_{10}$ ratio dropped to below 0.1 after 19:30 CST, further confirming that dust dominated the aerosol composition during this time.

From the evening of November 2nd to the early morning of November 3rd (19:30 to 06:00 CST), wind speeds in the near-surface layer (0-400 m) were observed to be within the range of 3-6 m/s, and the wind speeds above 500 meters significantly exceeded 10 m/s, predominantly originating from the northwest, accompanied by strong vertical mixing (Figure 1(d-f)). The range-corrected lidar return signal (RCS) indicated that the boundary layer mixing height was approximately 1.5 km. Thus, it is inferred that the accumulation and subsequent dissipation of local pollution during this period were jointly influenced by

long-range transported dust particles and local emissions.

On the evening of November 2nd, the near-surface $PM_{10}$ concentration began to plummet sharply. After 00:00 CST on November 3rd, the minimum value dropped below 10 μg/m³, while the proportion of $PM_{2.5}$ started to rise. Concurrently, the





RCS shows the aerosols plummeted, and the particle depolarization ratio below 2 km fell synchronously to less than 0.1, whereas a strong particle depolarization ratio (>0.3) persisted around 2 km.

It is noteworthy that around 03:00 CST on November 3rd, the intensity of the upper-level wind field began to weaken. This was specifically manifested by a decrease in the core wind speed, originally situated at an altitude of approximately 1500 meters (dropping to 9-12 m·s⁻¹), with its lower boundary descending to about 500 meters, forming a distinct wind speed gradient transition zone near 1000-1500 meters. Regarding wind direction evolution, the dominant wind direction in the 0-1000 meter atmospheric layer during the early part of this period (until around 02:30 CST) was northwesterly (NW). However, around 02:30 CST, a systematic shift in wind direction occurred in the atmospheric layer below 1000 meters: starting from the lower levels, the wind direction rapidly changed from northwesterly (NW) to northeasterly (NE), a transition that developed upwards from near the surface to 1000 meters. The boundary layer height was compressed close to the ground (<0.5 km), consistent with observations of weaker vertical exchange rates below 500 meters (Figure 1(e) vertical velocity). This is likely due to the change in near-surface temperature caused by the cold air from the north, which reduced vertical turbulence and thus inhibited the vertical development of the boundary layer (Seinfeld & Pandis, 2016). On the other hand, the northerly airflow played a crucial role in the horizontal transport of local pollutants, leading to a rapid decrease in overall aerosol concentration. With the wind shear and the weakening of vertical turbulence, the boundary layer lost its dust source, and the aerosol type shifted from dust-dominated to locally emitted fine particles. This phenomenon contradicts previous findings that a low boundary layer was the main cause of pollution accumulation, indicating that the factors causing the pollution event are likely jointly determined by the boundary layer and meteorological conditions (weakening wind speed, limited turbulent diffusion).

From the morning to the evening of November 3rd (06:00-18:00 CST), as the day progressed, the wind field exhibited significant temporal and spatial fluctuations. Near-surface wind speeds slightly increased compared to the night, with average wind speeds rising to 5-8 m·s⁻¹, and a localized instantaneous wind speed enhancement was observed to reach 9 m·s⁻¹ around 14:00 CST. Around noon at 12:00 CST, the previously continuous wind speed belt existing in the mid-atmospheric layer (1000-2000 m) began to split, forming a discontinuous two-layer structure. Concurrently, wind speeds above 2000 meters generally showed a weakening trend, leading to the complete dissipation of the strong upper-level wind core observed the previous night (>12 m·s⁻¹). The evolution of wind direction was also crucial; after shifting to the northeast during the night, the wind direction in the lower layer (0-1000 meters) consistently turned towards the south (S to SW) throughout the day. In contrast, within the 1000-2000 meter altitude range, a complex wind shear layer was observed, characterized by alternating westerly (W) and southwesterly (SW) winds. Above 2000 meters, the upper atmosphere essentially returned to a northwesterly (NW) wind. This inverse configuration of upper and lower wind directions (lower-level southerly winds and upper-level northwesterly winds) resulted in a significant anticyclonic vertical wind shear.

Starting at 12:00 CST on November 3rd, as daytime temperatures rose and the wind shear shifted predominantly to southerly winds, surface vertical turbulence intensified, and the boundary layer rapidly elevated to 1.5 km. Consequently, the overall aerosol concentration within the boundary layer began to increase. The particle depolarization ratio maintained at



around 0.2 below 1.5 km, indicating a relatively homogeneous mixing of aerosols within the boundary layer, primarily consisting of a mixture of dust and non-dust aerosols. We speculate that the non-dust aerosols mainly originate from anthropogenic emissions, while the dust may be derived from vertical mixing with residual dust aloft. However, after 18:00

on November 3rd, a distinct mixed dust layer (particle depolarization ratio of 0.1-0.2) appeared at around 1 km, suggesting that the dust could also be transported back from downwind areas previously carried away. This phenomenon is closely associated with changes in meteorological conditions, implying the redistribution of horizontally transported aerosols (such as regional pollution clusters) under the adjustment of boundary layer dynamic structures. Simultaneously, a stable aerosol layer with a particle depolarization ratio bigger than 0.3 persisted at the boundary layer top (1.5-2 km). The formation of this vertical

structure may be attributed to one primary factor: the continuous transport of dust aerosols by northwesterly winds at this altitude, reflecting the outcome of inter-regional transport.

From the evening of November 3rd to the early morning of November 4th (18:00-09:00 CST), during the night, near-surface wind speeds returned to a stable state, generally remaining below 5 m·s⁻¹. Within the atmospheric layer between 500-1500 meters, a wind speed belt with an intensity of 9-12 m·s⁻¹ was reestablished; however, its vertical coverage and duration

were significantly less than the similar structure observed on the night of November 2nd. At higher altitudes (2000-2500 m), a relatively strong wind core of 12-15 m·s⁻¹ reemerged. The wind direction exhibited vertical layering characteristics. The near-surface layer (0-500 m) was predominantly influenced by southeasterly (SE) winds, with local weak easterly winds interspersed. The atmospheric layer between 500-1000 meters was controlled by systematic southerly (S) winds, while the layer above 1000 meters was predominantly characterized by southwesterly (SW) winds, which gradually transitioned to

southerly winds by the end of the observation period. During this time frame, the boundary layer remained at a height of 1 km without significant diurnal variation, likely due to the wind shear at the 1 km level.

Throughout the entire observation period, the core of the westerly jet stream, which existed at an altitude of approximately 1000 meters with an intensity of 15 m·s⁻¹ on the night of November 2nd, significantly weakened and dissipated by the day on November 3rd. However, a westerly jet structure with adjusted intensity and position (around 2000-2500 meters, 12-15 m·s⁻¹)

was reestablished on the night of November 3rd. Regarding the near-surface wind field, the prevailing wind direction steadily shifted to southerly/southwesterly (S/SW) after 06:00 CST on November 3rd. This change in wind direction, coinciding with the typical timing following a cold front passage, indicates that the local influence of warm advection played a role after the cold front had passed. Lastly, a significant vertical discontinuity in the wind field consistently existed near 1000 meters, characterized by variable wind directions and strong vertical wind speed gradients at this altitude. The most intense shear event

occurred around 18:00 CST on November 3rd, with a wind speed difference exceeding 10 m·s⁻¹ observed within just 500 meters of vertical distance.

During the period of rapid pollutant clearance (before 2:00 CST on November 3rd), both the atmospheric boundary layer height and wind direction played crucial roles. The decrease in boundary layer height prevented the mixing of dust aerosols from above into the surface layer, and the strong northwesterly winds facilitated the rapid dispersion of anthropogenic



pollutants. Subsequently, the wind shear from northwest to southeast directions cut off the source of dust aerosols, while the

elevated boundary layer height also contributed to the dilution of anthropogenically emitted aerosols.

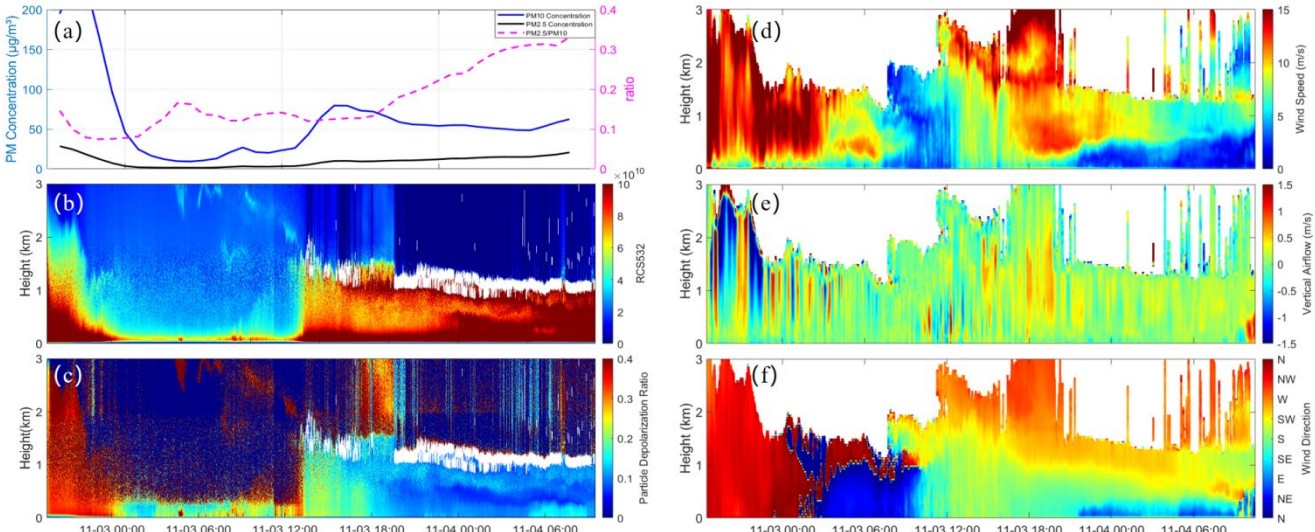

**Figure 1: In-situ and lidar observation results over Beijing from November 2, 2023, to November 4, 2023 (Beijing Time). (a) Near-surface PM$_{2.5}$ and PM$_{10}$ concentrations and their ratio from 00:00 on November 2, 2023, to 24:00 on November 4, 2023;**
**Observational results of the temporal and spatial distribution of the REAL and wind profiler lidar from 19:30 on November 2, 2023, to 09:30 on November 4, 2023, in the 0-3 km atmospheric layer over Beijing (b) 532 nm lidar range-corrected signal (RCS, representing vertical distribution information of relative aerosol loadings); (c) 532 nm particle depolarization ratio; (d) Temporal and spatial distribution of horizontal wind speed; (e) Temporal and spatial distribution of vertical velocity; (f) Temporal and spatial distribution of wind direction. The white areas in (b) and (c) represent cloud layers identified by the Vertical Distribution**
**Equalization (VDE) method, and the white areas in (d), (e) and (f) represent the absence of observation datasets.**

Figure 2(a) illustrates the spatio-temporal evolution of the total aerosol backscattering coefficient at 532 nm from 19:30

CST on November 2 to 09:30 CST on November 4, 2023. It reveals the complex vertical structural variations of aerosol

loadings during this pollution event: at the initial stage of pollution (evening of November 2 to early morning of November 3),

significant high-value (about $1\times10^{-5}$ m$^{-1}\cdot$sr$^{-1}$) appeared at low altitudes (0-1.5 km), indicating the accumulation of aerosols in

the near-surface layer; subsequently, with changes in meteorological conditions, a strong aerosol layer (backscattering

coefficient over $1\times10^{-5}$ m$^{-1}\cdot$sr$^{-1}$) emerged at higher altitudes (1.5-2.5 km), while the aerosol backscattering coefficient near-

surface layer is lower to $0.1\times10^{-5}$ m$^{-1}\cdot$sr$^{-1}$.



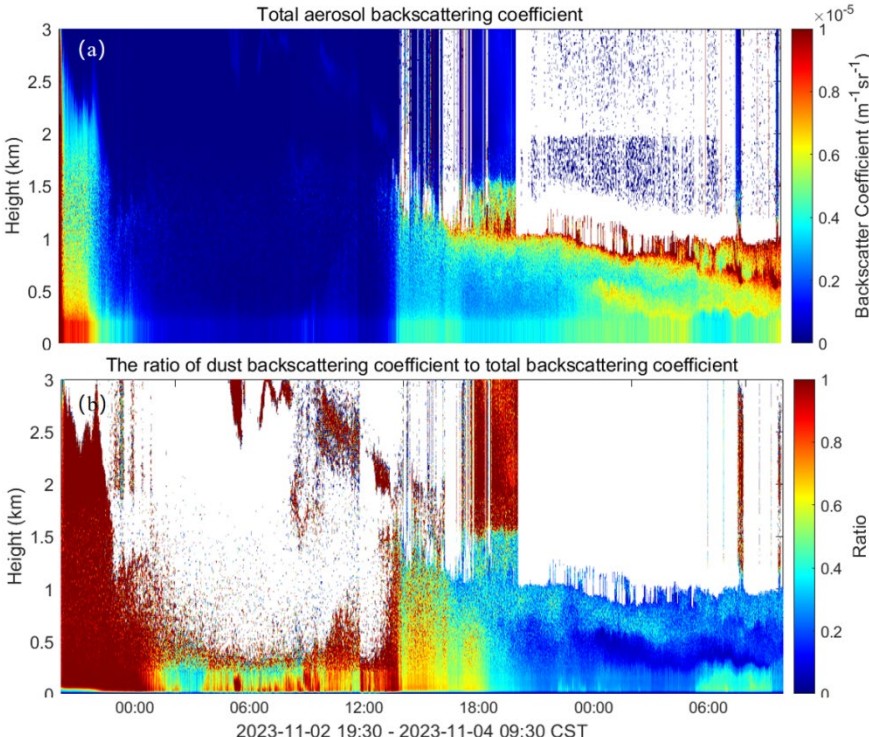

**Figure 2: (a) the spatial-temporal evolution of the total aerosol backscattering coefficient at 532 nm wavelength from 19:30 on**
**November 2, 2023, to 09:30 on November 4 (CST). (b) the ratio of dust backscattering coefficient to total backscattering coefficient.**
**The white areas indicate the absence information.**

To gain an in-depth understanding of the transport behavior of aerosols from different sources, we have separated dust
and non-dust aerosols based on the depolarization signal (Figure 2, b). It indicates that dust aerosols are predominantly located
at higher altitudes, while non-dust aerosols are mainly found at lower altitudes, and also indicates that the non-dust aerosols
mainly located in the lower atmospheric layer are predominantly by local anthropogenic emissions in the North China plain.
This classification enables us to investigate the transport fluxes of dust and non-dust aerosols separately.

To investigate the regional transport of aerosols, the temporal variations in PM concentrations in Beijing and its
surrounding cities during this period were analyzed (Figure 3). On November 2nd, the five cities simultaneously reached peak
$PM_{2.5}$ concentrations, indicating a regional fine particulate matter pollution event. The $PM_{2.5}/PM_{10}$ ratio exceeded 0.6 during
the peak period in all cities, confirming that the pollution was primarily composed of $PM_{2.5}$. Zhangjiakou first experienced an
explosive increase in $PM_{10}$ at 10:00 on November 2nd, followed by subsequent pollution peaks in Shijiazhuang, Tianjin, and
Chengde, revealing a dust transport pathway from northwest to southeast. Subsequently, both $PM_{10}$ and $PM_{2.5}$ concentrations
in the five cities showed a rapid decline, and the $PM_{2.5}/PM_{10}$ ratio also decreased to varying degrees, reflecting the effective
clearance of local pollutants by strong meteorological processes.



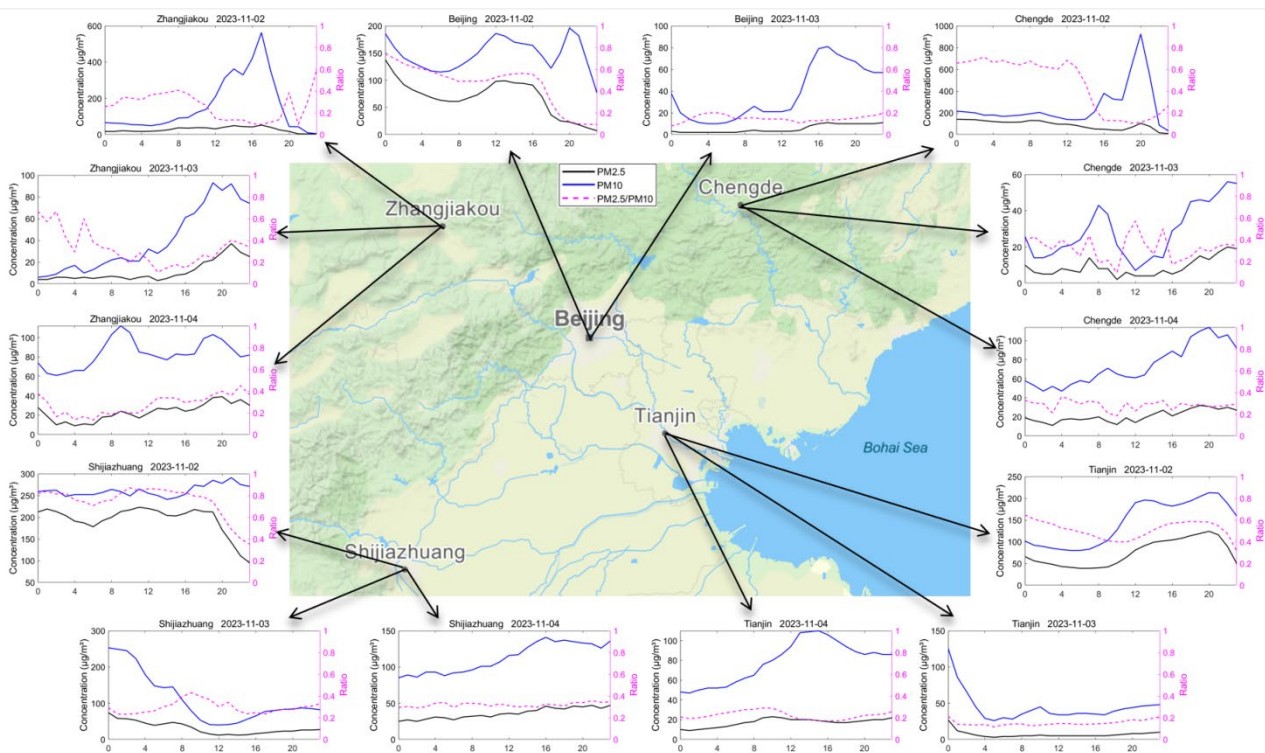


**Figure 3: The hourly air quality trends of major cities in the Beijing-Tianjin-Hebei region (Beijing, Tianjin, Zhangjiakou, Chengde) from November 2 to 4, 2023 are presented. The central map (© Mapbox, available at https://www.mapbox.com/mapbox-studio) illustrates the geographical distribution of the cities, with surrounding subplots showing the daily variation patterns of PM$_{2.5}$ (black line), PM$_{10}$ (blue line) concentrations, and the PM$_{2.5}$/PM$_{10}$ ratio (pink dashed line) for each city. The concentration units are micrograms per cubic meter (μg/m³), and the time span is from 0:00 CST to 23:00 CST daily.**






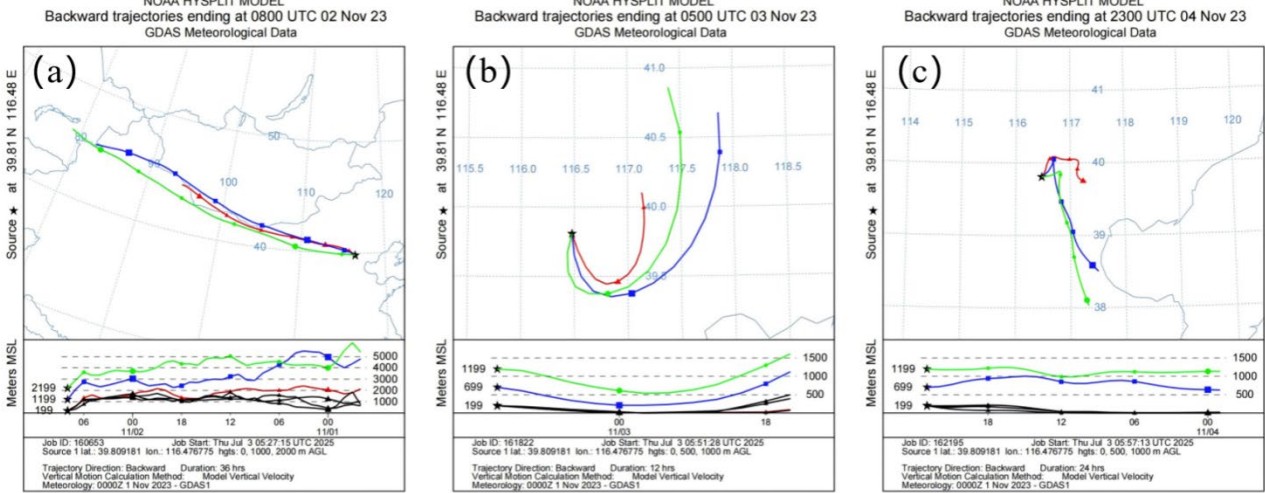

**Figure 4: HYSPLIT modeled 48 h airmass backward trajectories for Beijing on 02-04 November 2023. (a) HYSPLIT modeled 36 h**
**on 02 November 2023 at 00:00 China Standard Time (CST) (UTC+8). All altitudes (green, blue, red and black), the trajectories mainly originated above the Taklimakan Desert. (b) HYSPLIT modeled 12 h on 02 November 2023 at 21:00 CST (UTC+8). All altitudes (green, blue, red and black), the trajectories mainly originated above the North China Plain. (c) HYSPLIT modeled 24 h on 04 November 2023 at 15:00 CST (UTC+8). At higher altitudes (green), the trajectories mainly originated above the North China Plain, while at lower altitudes (blue and red) they originated above the Taklimakan Desert.**

Figure 4 shows the backward trajectories consistent with the wind direction results from the HYSPLIT model and wind profiler lidar observations, revealing the sources of aerosols within the aerosol layer during the air pollution event in Beijing, November 2023. From Figure 4(a) (16:00 CST on November 2), it can be observed that the air mass trajectory follows a northwest-southeast path, originating from southern Mongolia and western Inner Mongolia, China (39°~41°N, 114°~116°E), indicating that the air mass during this period was primarily transported from the northwest, consistent with the
contemporaneous wind field characterized by "predominantly northwesterly winds." In Figure 4(b) (Beijing Time, 13:00 on November 3), there is a dramatic change in the trajectory pattern, with all altitude levels turning southwestward. The air mass below 1000 meters originates from the southern North China Plain (35°N, 115°E), while the 1500-3000 meter layer recirculates inland from the Yellow Sea (34°N, 122°E), reflecting the warm and moist air mass along the coastal return flow. The trajectory at 500 meters altitude passed over the Bohai Bay (38°N, 120°E) 12 hours prior, suggesting a relatively clean air mass, although
the possibility of sea salt particles being mixed in cannot be ruled out. In Figure 4(c) (07:00 CST on November 5), the trajectories at all altitudes exhibit a southeast-northwest recirculation trend, with the lower layer (199~699 m) air mass originating from the eastern North China Plain (38°~40°N, 117°~119°E), driven northwestward by southerly winds. The upper layer (1500~2199 m) trajectories extend southwestward, indicating the reverse transport after the recovery of the northwesterly winds at high altitudes. Combined with the observations of particle depolarization, it can be confirmed that dust aerosols
primarily originate from the northwest (Taklamakan Desert), while non-dust aerosols mainly come from anthropogenic emission sources in the North China region.



To quantitatively describe the aerosol transport flux during this event, we obtained the horizontal and vertical fluxes of aerosols by combining the estimated dust/non-dust aerosol mass concentrations with wind profile data (as shown in Figure 5). The horizontal flux of dust aerosols (Figure 5, a) was predominantly positive during the evening of November 2nd, approximately 5000 $\mu g \cdot m^{-2} \cdot s^{-1}$, indicating transport towards the southeast, consistent with the northwesterly wind dominance at that time. Around the early morning of November 3rd, the horizontal flux of dust aerosols sharply dropped from a value of lower than 2000 $\mu g \cdot m^{-2} \cdot s^{-1}$ to near zero. A similar phenomenon was found for the horizontal flux of non-dust aerosol (Figure 5, c), while only the significant values of flux are mainly in the lower layers (below 1 km). Notably, the low-level wind shear change at 02:30 CST on November 3rd (from northwest to northeast) led to a reversal of the dust horizontal flux from positive to near zero within an hour, directly cutting off the dust supply from the source region and causing a sharp decrease in near-surface $PM_{10}$ concentrations. This phenomenon confirms the modulating effect of wind shear on pollutant clearance proposed by Zhang et al. (2020), but this study further reveals its impact on the transformation mechanism of aerosol types. Around 12:00 CST on November 3rd, due to another abrupt change in wind direction, the horizontal flux of dust aerosols also dramatically shifted to a negetive value of about -5000 $\mu g \cdot m^{-2} \cdot s^{-1}$ below 2 km. The horizontal flux of non-dust aerosols was relatively weak during the earlier period and stabilized before noon on November 3rd, with an intensity weaker than that of dust, but showed a clear directional reversal after noon on November 3rd. We deduce the dust transported to the downstream region was probably blown back as the wind shifted from northwesterly to southeasterly, which indicates the aerosol transport mechanism in a larger region. Strong upward transport of dust aerosols was observed during the evening of November 2nd, with the vertical flux peaking over 5 $\mu g \cdot m^{-2} \cdot s^{-1}$, corresponding to strong turbulent conditions (Figure 5, b); early morning on November 3rd, it shifted to significant deposition, with a vertical flux of about -4 $\mu g \cdot m^{-2} \cdot s^{-1}$, reflecting dust deposition caused by boundary layer compression and a second upward flux of over 5 $\mu g \cdot m^{-2} \cdot s^{-1}$ appeared in the afternoon on November 3rd, indicating the secondary lifting effect due to boundary layer elevation. The vertical flux of non-dust aerosols consistently remained slightly positive (0-3 $\mu g \cdot m^{-2} \cdot s^{-1}$), with only a deposition flux observed at 14:00 CST on November 3rd at an altitude of 800 meters, suggesting that fine particles mainly diffused upwards in the vertical direction, contrasting sharply with the intermittent deposition of dust.

The most notable dust/non-dust aerosol flux events in Figure 5 occurred during the evening of November 2nd to the early morning of November 3rd, as well as from the morning to noon on November 3rd. These periods correspond to the strong horizontal wind speed events and wind direction abrupt changes depicted in Figure 1. Strong horizontal winds and strong boundary layer turbulence mixing are the primary driving forces behind the lifting of aerosol from the ground and the transport of mixture dust and anthropogenic emission particles.

Compared to the study of near-surface aerosol fluxes, the vertical cross-sectional aerosol fluxes in this research are particularly useful for understanding the inter-regional transport of aerosols. However, if similar lidar devices were deployed in multiple cities across the Beijing-Tianjin-Hebei region, it would be possible to further quantitatively analyze the mutual transport of aerosols between cities. Regrettably, due to the limitations in observation duration, this study was unable to utilize




long-term observational data to investigate pollution processes and pollutant transport characteristics under various weather patterns.

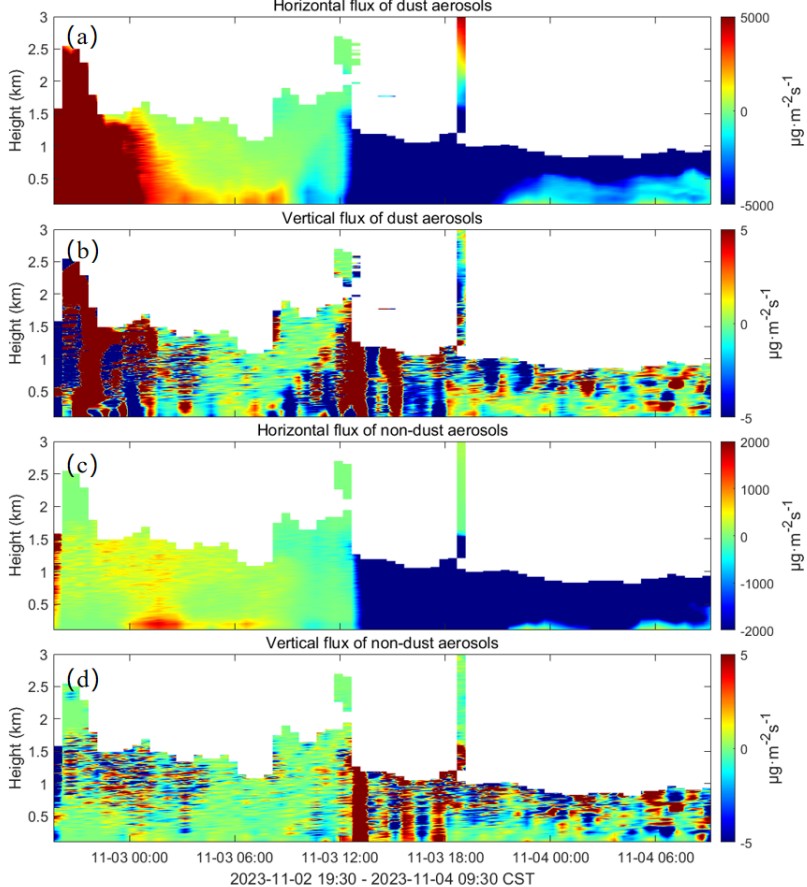

**Figure 5: Aerosol flux results from 19:30 on November 2, 2023, to 09:30 on November 4, 2023 (China Standard Time, CST), in the 0.1-3 km atmospheric layer over Beijing. (a) Horizontal flux of dust aerosols; (b) Vertical flux of dust aerosols; (c) Horizontal flux of non-dust aerosols; (d) Vertical flux of non-dust aerosols.**


Figure 6 illustrates the spatial-temporal distribution of emission rates of dust and non-dust aerosols in the horizontal and vertical directions over the Beijing region from 19:30 CST on November 2 to 09:30 CST on November 4, 2023. The horizontal emission rate of dust aerosols (Figure 6, a) fluctuated significantly due to abrupt changes in the wind field. In the evening of November 2, driven by strong northwesterly winds, the horizontal emission rate showed a notable positive value

(approximately 8-10 m·s⁻¹), indicating transport of dust towards the southeast. Early on November 3, with the abrupt shift of the low-level wind direction to northeasterly, the horizontal emission rate sharply reversed to a strong negative value (< -0.01 m·s⁻¹), signifying a sudden increase in the efficiency of transporting dust northwestward. This drastic reversal in emission rates directly explains the rapid changes in the distribution of the high particle depolarization ratio layer observed in Figure 1, (c), as well as the abrupt decrease in near-surface PM₁₀ concentrations—strong transport efficiency rapidly removed local dust.



The horizontal emission rate of non-dust aerosols (Figure 6, c) followed a general trend consistent with that of dust, indicating that its horizontal transport was also controlled by the wind field.

In the evening of November 2, the boundary layer height was approximately 1.5 km, with strong convection, and the vertical emission rate of dust aerosols exceeded 0.01 m·s⁻¹, indicating that strong convection efficiently lifted dust. Early in the morning of November 3, the boundary layer height compressed to less than 0.5 km, with weaker convection, and the
vertical emission rate shifted to around -0.01 m·s⁻¹, reflecting a dominant deposition process. In the afternoon, as the boundary layer rose again, a peak in positive emission rates reemerged.

The most notable characteristic of the vertical emission rate of non-dust aerosols (Figure 6, d) was that it was mostly positive (0 to 0.02 m·s⁻¹) throughout the observation period. This suggests that for non-dust aerosols, the vertical diffusion rate caused by convection was continuously greater than their minimal gravitational settling rate, resulting in a net upward transport.
This observation dynamically explains the slower decrease in $PM_{2.5}$ concentrations compared to $PM_{10}$ in Figure 1, (a) and the phenomenon of pollution lingering, as fine particles, due to their extremely slow settling speed, are difficult to effectively remove through dry deposition and are more likely to be entrained and remain in the atmosphere for extended periods.

The results indicate a significant regulatory effect of meteorological conditions on aerosol emission rates: the diurnal variation in planetary boundary layer height drives the alternation between positive and negative vertical emission rates;
whereas abrupt changes in wind direction modulate the intensity of horizontal emission rates by altering the transport pathways.





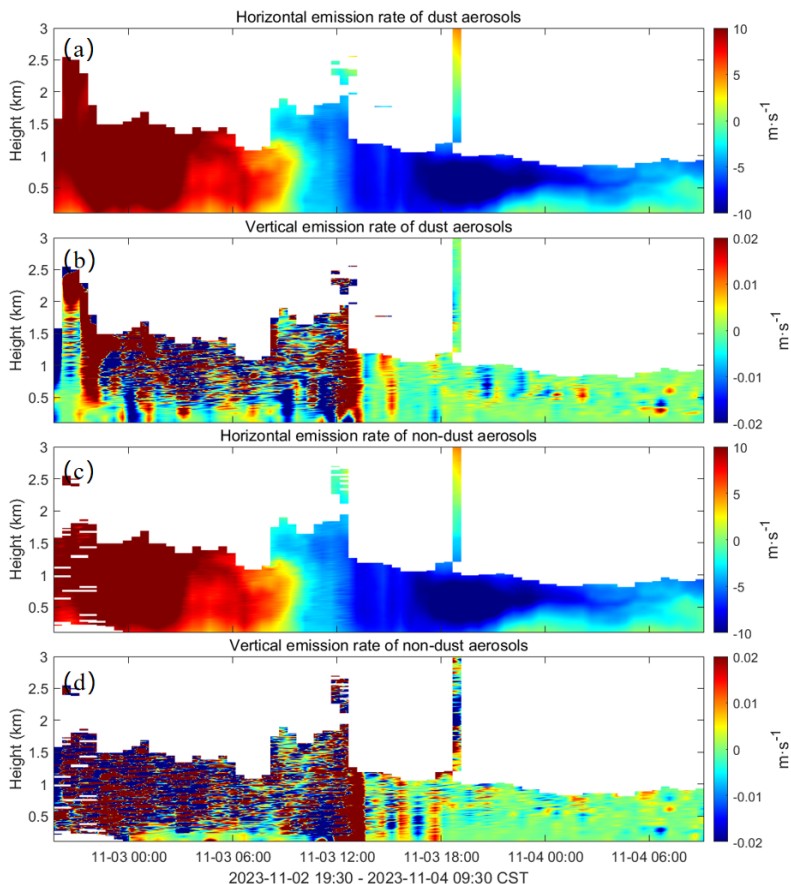

**Figure 6: Aerosol emission rate results from 19:30 on November 2, 2023, to 09:30 on November 4, 2023 (CST), in the 0.1-3 km atmospheric layer over Beijing. (a) Horizontal emission rate of dust aerosols; (b) Vertical emission rate of dust aerosols; (c) Horizontal emission rate of non-dust aerosols; (d) Vertical emission rate of non-dust aerosols.**





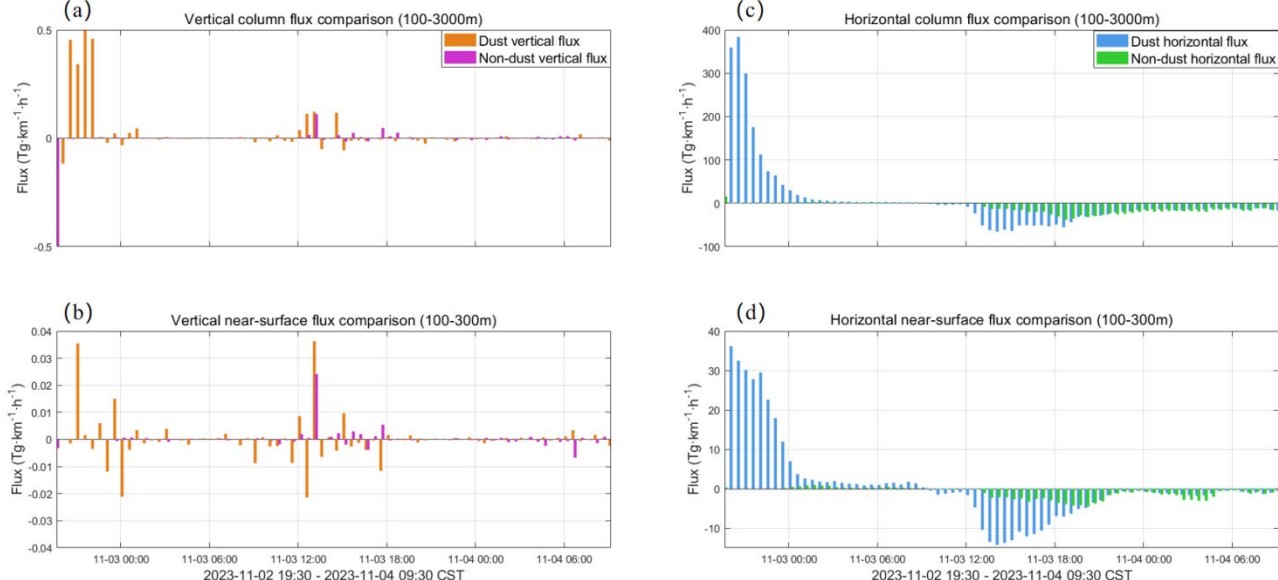


**Figure 7: Distribution of aerosol column transport fluxes from November 2 to November 4, 2023. (a) Vertical column transport fluxes of aerosols within the atmospheric layer of 100-3000 meters; (b) Horizontal column transport fluxes of aerosols within the atmospheric layer of 100-3000 meters; (c) Vertical near-surface column transport fluxes of aerosols within the atmospheric layer of 100-300 meters; (d) Horizontal near-surface column transport fluxes of aerosols within the atmospheric layer of 100-300 meters.**

To analyze the changes in the near-surface and total fluxes of dust and non-dust aerosols during the diffusion process of this event, we obtained the column transport fluxes from 100-3000 meters (approximating the aerosol content throughout the entire atmospheric layer) and from 100-300 meters (representing the ground-level aerosol content) (Figure 7). Negative non-dust fluxes can be used to characterize the meteorological conditions' ability to clear locally emitted anthropogenic aerosols; positive non-dust fluxes indicate the meteorological conditions' capacity to accumulate locally emitted anthropogenic aerosols.

Negative dust fluxes can be used to characterize the meteorological conditions' ability to clear already accumulated dust aerosols locally; positive dust fluxes indicate the meteorological conditions' capacity to accumulate dust aerosols transported to the local area. The columnar fluxes from 100-300 meters show that the vertical positive fluxes of non-dust are generally an order of magnitude higher than those of dust, on the contrary, the vertical negative fluxes of non-dust are generally lower than those of dust, which indicates that dust primarily originates from long-range transport and non-dust are emitted from locally.

Examining the column transport fluxes from 100-3000 meters, the highest value of positive vertical fluxes are about 0.5 Tg·km$^{-1}$·h$^{-1}$ for dust aerosols and 0.15 Tg·km$^{-1}$·h$^{-1}$ for non-dust aerosols, while the highest value of negative vertical fluxes are about 0.15 Tg·km$^{-1}$·h$^{-1}$ for dust aerosols and 0.5 Tg·km$^{-1}$·h$^{-1}$ for non-dust aerosols. It is observed that both dust and non-dust aerosol horizontal fluxes exceed their respective vertical fluxes, indicating that the horizontal dispersion of pollutants is more pronounced than their vertical dispersion. For the Southeastward horizontal transport (positive values shown in Figure 7, c), the maximal total column dust horizontal fluxes (100-3000 m) are estimated to be about 380 Tg·km$^{-1}$·h$^{-1}$. In comparison, the maximal total column non-dust horizontal fluxes are found about 10 times smaller (about 30 Tg·km$^{-1}$·h$^{-1}$). Regarding the





near-surface level (Figure 7, d), the maximal column dust horizontal fluxes (100-300 m) are estimated to be about 36 Tg·km⁻¹·h⁻¹. In comparison, the maximal column non-dust horizontal fluxes are found about 8 times smaller (about 3.5 Tg·km⁻¹·h⁻¹). For the Northwestward horizontal transport (negative values shown in Figure 7, c), the maximal total column dust horizontal fluxes (100-3000 m) are estimated to be about 60 Tg·km⁻¹·h⁻¹. In comparison, the maximal total column non-dust horizontal fluxes are found about 1-2 times smaller (about 40 Tg·km⁻¹·h⁻¹). Compared with Southeastward horizontal transport, the Northwestward dust aerosols flux decreased significantly (approximately one-sixth of the Southeastward flux), while the non-dust aerosols flux slightly increased (about 1.3 times that of the southeastward flux). It may indicate that the long-range transported dust was mixed with local emissions in the downstream region of Beijing then impact the PM concentrations of Beijing. And also, it can indicate that the transportation cross-section in Southeastward is smaller than that in northwestward, which agrees with the terrain around Beijing region (Beijing has mountains in its northwest and plains in its southeast). In addition, to confirm with the near surface PM concentration over five cities (Figure 4), it seems the aerosol transport cross-section probably reaches over 100 km levels. Regarding the near-surface level, the maximal column dust horizontal fluxes (100-300 m) are estimated to be about 14 Tg·km⁻¹·h⁻¹. In comparison, the maximal column non-dust horizontal fluxes are found about 3 times smaller (about 4.5 Tg·km⁻¹·h⁻¹). Compared with Southeastward horizontal transport, it seems Northwestward horizontal transport brought 4 times less dust aerosols back but 1.2 times more non-dust aerosols. It indicates that the impact of local emissions is more than dust on the near surface. In general, a comparison between southeastward and northwestward air flows suggests that air pollutants over Beijing are more easily influenced by northwestward flows, which can bring about twice as much aerosol loading from the southeast region to Beijing.

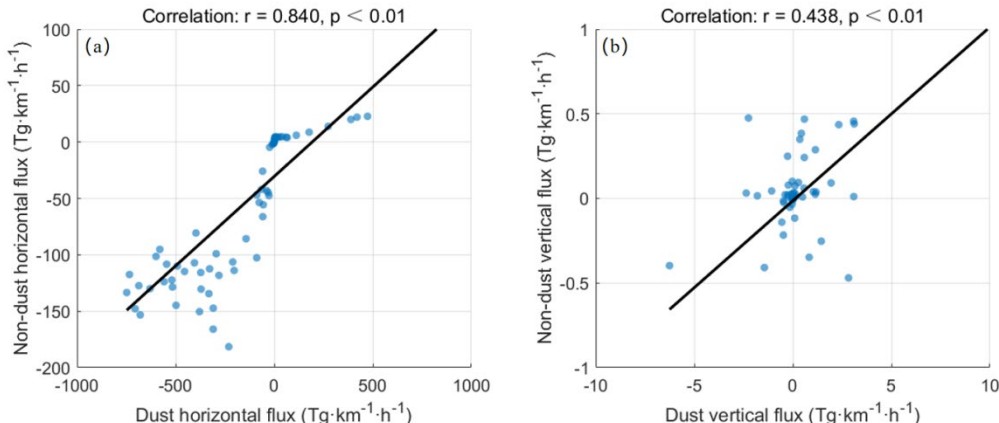

**Figure 8: Correlation between Dust and Non-Dust Aerosol Fluxes. (a) Scatter plot showing the correlation between dust horizontal flux and non-dust horizontal flux. (b) Scatter plot illustrating the correlation between dust vertical flux and non-dust vertical flux.**

The correlation analysis of aerosol fluxes reveals significant differences in the transport mechanisms of dust and non-dust components (Figure 8). The horizontal fluxes exhibit a high positive correlation (r = 0.856, p < 0.01), indicating that dust and non-dust aerosols are controlled by the same atmospheric dynamical processes in the horizontal direction. In contrast, although the vertical fluxes show a positive correlation trend, the correlation is significantly weaker (r = 0.438, p < 0.01), suggesting a



big difference in their vertical transport processes. It is probably because one mainly comes from surface emission, while the other is mainly from sedimentation, and their emission rates are also comparatively different.

## 4 Discussion and Conclusions

A strong dust transport event occurred from North-West China to the North China Plain in November 2023. During the transport periods, the Southeast-toward migration of cold air-masses carries not only the long-range transported dust but also local anthropogenic pollutants from the North China Plain to the Southeast region. This study quantitatively analyzed the vertical-horizontal flux structure of aerosols in the Beijing area using the polarization lidar and Doppler wind lidar observations on 2 November 19:00 CST to 4 November 9:30 CST, 2023. The main research findings are summarized as follows:

1. Meteorological factors exert a significant regulatory effect on aerosol transmission

Temporal and spatial variations in wind direction and speed are the core drivers of horizontal aerosol transmission. From the evening of November 2 to the early morning of November 3, the low-altitude wind direction in Beijing shifted from Northwest to northeast, causing the horizontal transmission direction of dust aerosols to change from southeastward to northwestward. This confirms the path rule that "northwesterly winds push dust to spread southeastward, while northeasterly winds drive dust to transmit northwestward". Meanwhile, the dissipation and reconstruction of the upper-level westerly jet further affected the vertical transmission range of aerosols. In addition, the diurnal variation of the planetary boundary layer (PBL) height governs the vertical diffusion of aerosols by regulating turbulence intensity. When the boundary layer is stable at night, dust tends to settle and accumulate; during the day, the rising boundary layer, accompanied by strong turbulence, promotes the upward diffusion of aerosols, which is highly synchronized with the positive-negative transition of vertical emission rate (positive during uplift and negative during sedimentation).

2. There are fundamental differences in emission and diffusion characteristics between dust and non-dust aerosols.

The emission rate of dust aerosols fluctuates drastically, with their horizontal transmission significantly affected by sudden wind direction changes. Vertically, they exhibit an alternating "uplift-sedimentation" pattern and are mainly concentrated in the near-surface layer below 1000 meters, reflecting their close association with natural/anthropogenic activities such as surface dust emission. In contrast, non-dust aerosols (dominated by anthropogenically emitted fine particles) have a stable and low-intensity emission rate. Their horizontal transmission trend is consistent with that of dust but responds more gently. Vertically, they primarily undergo weak upward diffusion, with rare large-scale sedimentation—characteristics that reflect the continuous emission from anthropogenic sources (e.g., industry and transportation). Moreover, due to their small particle size and slow sedimentation rate, fine particles are more likely to accumulate within the boundary layer.

3. The effectiveness of multi-source observation technologies and flux calculation methods is verified.

In this study, Real lidar was used to retrieve aerosol optical properties (e.g., particle depolarization ratio) for accurate classification of dust and non-dust aerosols. Combined with Doppler wind lidar to obtain high-resolution wind profiles, the



eddy covariance method was applied to calculate aerosol mass fluxes. This approach successfully captured the temporal and
spatial distribution of emission rates of the two types of aerosols in horizontal and vertical directions. This technical system
can effectively analyze the coupling mechanism of "meteorological conditions—pollution sources—diffusion processes" and
provides a reliable method for quantifying anthropogenic aerosol emission rates in cities.

4. The synergistic impact of regional pollution transmission and local emissions is significant.

Observations of PM concentrations in cities around Beijing showed that on November 2, Zhangjiakou was the first to
experience an explosive increase in $PM_{10}$ concentration, followed by pollution peaks in Shijiazhuang, Tianjin and other cities.
This reveals the cross-regional transmission path of dust from northwest to southeast. Additionally, the peak $PM_{2.5}$
concentration in Beijing corresponds to the stable emission of non-dust aerosols, indicating that the superposition of regional
transmission and local anthropogenic emissions is a key cause of this pollution event. Severe weather processes (e.g., cold air
advection) have a significant scavenging effect on both types of pollutants, further verifying the potential of meteorological
intervention in pollution control. The total amount of dust transmission rate is estimated to be about 1360Tg·km$^{-1}$·h$^{-1}$ and the
total amount of local pollutants transmission rate is estimated to be about 770 Tg·km$^{-1}$·h$^{-1}$ for the entire troposphere.

In conclusion, the established polarization-wind field joint inversion framework in this study overcomes the spatial
representativeness issues of traditional single-point flux observations. The aerosol type separation achieved by the
depolarization ratio method enables the quantitative decomposition of the contributions to the aerosol flux. Combining both
vertical and horizontal fluxes of aerosol can be used to estimate the total transport amount of dust along its trajectories and
also the total transport amount between the cities. The aerosol type-specific flux observation technology developed in this
study provides new insights for joint pollution prevention and control in urban agglomerations like the Beijing-Tianjin-Hebei
region and similar city clusters. This holds important reference value for the implementation of cross-regional coordinated air
pollution control in the context of the integrated development of the Beijing-Tianjin-Hebei and Yangtze River Delta regions.
It proves that using lidar remote sensing to study aerosol fluxes under different meteorological conditions (wind direction,
wind speed and boundary layer height) provides data suitable for a scientific basis for urban air pollution early warning and
the formulation of regional joint prevention and control strategies (e.g., northwest source control for dust and local emission
reduction for non-dust aerosols). Furthermore, the aerosol fluxes profiles can be useful to estimate the amount of pollutant
across-region transport. However, there are limitations: the observation duration is limited (38 hours), making it difficult to
cover fluxes characteristics under various weather patterns; the mass concentration conversion factor uses fixed values without
considering the hygroscopic growth effect during aerosol aging; and the non-dust contributions are assumed to be
anthropogenic emissions, of which the sources need to be further confirmed and a more accurate mass concentration conversion
factor is required. Moreover, the horizontal flux calculation does not incorporate multi-city network observations, limiting the
closure verification of regional transport fluxes. In the future, lidar observation networks can be deployed to track regional
transport fluxes in real-time and optimize emergency emission reduction strategies for heavy pollution weather. For instance,
when horizontal flux monitoring indicates an increase in dust input, dust suppression measures at construction sites can be



initiated in advance; whereas when the vertical flux of non-dust aerosols remains positive, it is necessary to strengthen the control of industrial point sources and mobile sources.

*Code and data availability*. Code and data can be provided by the corresponding author on request.

*Author contributions*. All the authors made contributions to this research work and manuscript. In particular, LW, ZY, YH and XW designed the whole strategy of this work. LW organized the observation campaign. ZT analyzed the data and wrote the manuscript draft. SL, CY participated the data analysis. LW, ZY, YH, BZ, DC and XW participated in the scientific discussions,
reviewed and proofread the manuscript. TL participated in the observations and data collection. LL, BZ, DC, ZB, YC and XW reviewed and proofread the manuscript. LW, ZY, YH, LL, XW and YC acquired the research funding and led the study. All authors have read and agreed to the published version of the manuscript.

*Acknowledgements*. This work was supported by the National Natural Science Foundation of China (grant Nos. 42475147,
62105248, 42575141 and 42575203).

*Competing interests*. The authors declare no conflict of interest.

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
