# Peer review of "Investigation of Aerosol Transport Flux Structure over Beijing Based on Lidar Observations and the Impact of Dust Transport on Air Quality"

_EGUsphere, 2025_

## Author Comment (AC2)

**Response to the Reviewer's Comments 2**

January 28, 2026

The authors would like to thank the reviewer for the careful handling and valuable guidance throughout the review process, which have played a crucial role in polishing the paper and ensuring its academic rigor. We have fully addressed all the requirements and feedback put forward by the editor. They are listed one by one in this letter and corresponding revisions have been made in the manuscript, marked with blue text.

**Major comments:**

1. The paper focuses on a single haze event to investigate aerosol fluxes. This limitation has been recognized by the authors (e.g., line 570), however, I would recommend to discuss the reasons behind this choice in a greater detail, otherwise the study appears to be only of local interest.

**Answer:** We sincerely appreciate the valuable comments provided by the reviewers. The similar concern was also pointed out by the reviewer 1. We agree that investigating multiple events would offer a more comprehensive perspective. However, we intentionally selected this specific 38-hour event (from 19:30 on November 2 to 09:30 on November 4, 2023) for an in-depth case study. Because it represents a complete and typical pollution cycle driven by the complex interplay of meteorological factors and boundary layer dynamics, which constitutes the core focus of our research. This case is possible to elucidate the fundamental mechanisms governing aerosol transport under the typical monsoon during winter over the North China Plain (NCP). Moreover, similar studies such as Huang et al., 2024 has demonstrated that a single case study is also possible to investigate the aerosol transport mechanisms in detail. In addition, we are also considering to conduct a long-term observation after this report in future.

The detailed reasons for selecting this case are as follows, which are summarized from manuscript:

This event encompasses the entire lifecycle of a pollution episode, from pollutant accumulation to dissipation, driven by distinctive and rapidly evolving meteorological conditions. As illustrated in Figure 1, this period is characterized by the following features: a strong northwesterly wind event associated with cold air advection initially transported dust aerosols to Beijing (evidenced by a high particle depolarization ratio > 0.3, Figure 1c), followed by rapid pollutant scavenging. A notable low-level wind shear occurred around 02:30 on November 3, when the near-surface wind direction abruptly shifted from northwesterly (NW) to northeasterly (NE) (Figure 1f). This wind shear significantly altered the direction of horizontal aerosol fluxes (Figures 5a, c), effectively cutting off the dust source and leading to a sharp decline in near-surface $PM_{10}$ concentrations. This demonstrates how large-scale synoptic variations modulate local air quality.

There were pronounced diurnal variations in the planetary boundary layer height (Figures 1b, e). Nocturnal boundary layer compression to < 0.5 km facilitated aerosol sedimentation (negative vertical fluxes, Figure 5b). Conversely, daytime boundary layer development to 1.5 km enhanced vertical turbulent mixing and upward aerosol transport (positive vertical fluxes). This enabled us to quantitatively analyze the role of boundary layer dynamics in aerosol diffusion.

The event has also exhibited distinct regional characteristics. As shown in Figure 3, pollution peaks sequentially occurred in cities from Zhangjiakou (northwest) to Beijing and Tianjin (southeast), revealing a classic regional transport pathway. Backward trajectories simulated by the HYSPLIT model (Figure 4) further confirmed that air masses originated from dust source regions in the northwest and the North China Plain with intense anthropogenic emissions.

Therefore, despite a case study, it reveals the synergistic impacts of regional transport and local emissions, a common challenge faced by urban agglomerations such as Beijing-Tianjin-Hebei. (Line 133-137 were revised for clarify in the manuscript)

2. There is a major assumption that all non-dust aerosol is of anthropogenic origin. Already from the abstract the authors state that "the vertical and horizontal fluxes profiles of two type aerosols are calculated" (lines 18-19), referring to dust and non-dust aerosol (line 17). This is misleading for several reasons. To begin with, dust is an aerosol type but non-dust is a category. In addition, non-dust aerosol can also include natural aerosol, such as sea salt or volcanic aerosol. The authors recognize this assumption but only at the conclusions (lines 572-574). The assumption and implications should be already discussed before the results.

**Answer:** We sincerely appreciate the valuable comments provided by the reviewers. We agree that this clarification is crucial and should be introduced earlier in the manuscript. Within the specific context of our study area, Beijing—a megacity situated in the North China Plain, far from any oceans or active volcanoes—the major natural non-dust aerosol sources (sea salt and volcanic emissions) are negligible, in particular within the lower troposphere. So we can conclude the dominant sources of non-dust aerosols in this region are mainly anthropogenic, stemming from activities such as industrial emissions, vehicle exhaust, and residential combustion (Zhang et al., 2013; Huang et al., 2014; Liu et al., 2017).

In response to this comment, we have revised the manuscript to explicitly state and justify this regional assumption in the Instrumentation and Methodology section (Section 2 lines 207 -212 in the revised manuscript), where aerosol type classification is described. Additional text has been included to clarify that equating "non-dust aerosols" with "anthropogenic aerosols" constitutes a reasonable simplification based on the specific spatio-temporal context of this observation event (Beijing, winter 2023) and the regional environmental characteristics (an inland city with high-intensity anthropogenic emissions). This simplification allows us to focus on investigating the differences in transport fluxes between the two aerosol types that exert a decisive influence on Beijing's air quality: long-range transported dust and local/regional anthropogenic emissions.

3. The results section is highly descriptive, meaning that the authors describe the specific temporal and spatial evolution of the 38h pollution event in Beijing in a great detail. In addition, the description goes from parameter to parameter rather than following a day/time narration. I suggest to revise and shorten the descriptive parts.

**Answer:** We appreciate the reviewer's insightful and constructive comment on optimizing the structure and conciseness of the Results section. In response, we have revised the Results section comprehensively following the suggestions. Specifically, we reorganized the narrative framework to follow a clear time sequence centered on the 38-hour pollution event in Beijing, instead of describing parameters separately. We have also streamlined and optimized the descriptive content throughout the section, with focused refinement on the interpretation of Figure 1 and Figure 5.

The revised content now presents the evolution of the pollution event in three distinct phases, corresponding to the key meteorological transitions observed. This structure effectively highlights the dynamic relationships between aerosol properties, wind field changes, and flux variations, while removing redundant details to enhance the overall clarity and readability of the results (line 271-298 and line 379-399 in the revised ).

4. Figure 1 needs improvements. Please provide the figure in better quality and enlarge the fonts as the axis are very hard to read. It seems to me that the axis start is at 11-02 at 18:00 local time. However, in line 241 authors refer to measurements on the 00:00 CST on November 2nd, which is not shown in the figure. Also, the PM2.5/PM10 ratio of 0.8 (line 242) is not shown in Fig.1a. Is this intentional? If so, please state it.

**Answer:** We thank the reviewer for the careful review and valuable suggestions regarding Figure 1. In response, we have provided a higher-resolution vector version of the figure with enlarged axis labels to improve readability.

Regarding the time axis, the figure indeed starts at 19:30 CST on November 2, 2023, as consistently indicated in the captions and descriptions. The reference to measurements at 00:00 CST on November 2 and the $PM_{2.5}/PM_{10}$ ratio of 0.8 mentioned in the original manuscript were based on data from Beijing's city-level monitoring (shown in Figure 3 for November 2), not the lidar observation period displayed in Figure 1. To avoid confusion, we have revised the corresponding description in the Results section to clarify the data source and remove ambiguous statements.

Thank you again for your helpful comments.

[Figure]

**Figure 1: In-situ and lidar observation results over Beijing from 19:30 on November 2, 2023, to 09:30 on November 4, 2023 (CST). (a) Near-surface PM_{2.5} and PM_{10} concentrations and their ratio from 19:30 on November 2, 2023, to 09:30 on November 4, 2023; Observational results of the temporal and spatial distribution of the REAL and wind profiler lidar from 19:30 on November 2, 2023, to 09:30 on November 4, 2023, in the 0-3 km atmospheric layer over Beijing (b) 532 nm lidar range-corrected signal (RCS, representing vertical distribution information of relative aerosol loadings);**

(c) 532 nm particle depolarization ratio; (d) Temporal and spatial distribution of horizontal wind speed; (e) Temporal and spatial distribution of vertical velocity; (f) Temporal and spatial distribution of wind direction. The white areas in (b) and (c) represent cloud layers identified by the Vertical Distribution Equalization (VDE) method, and the white areas in (d), (e) and (f) represent the absence of observation datasets.

**Minor comments:**

1. In the introduction I missed literature references such as Wandinger et al. 2004.

**Answer:** We have conducted a thorough literature search and supplemented the relevant references, including Wandinger et al. (2004), in the Introduction section to strengthen the contextualization of our research and better highlight the advancements of the present study against prior work (Line 115 in the revised).

*The pioneering work by Wandinger et al. (2004) successfully demonstrated the application of the eddy covariance technique by combining Doppler wind lidar and Raman aerosol lidar to retrieve vertical aerosol flux profiles, validating the core methodology of synergistic lidar observations (Wandinger et al., 2004).*

2. Line 47: replace "propagation" with "transport" or "transfer".

**Answer:** The manuscript was modified accordingly (Line 47 in the revised).

*This vertical transport mechanism is crucial for the initial step of regional-to-intercontinental transport.*

3. Lines 236-239: Please introduce Fig.1 fully, before discussing the different subplots.

**Answer:** We have revised the manuscript by adding a comprehensive introductory description of Figure 1 prior to the discussion of its subplots (Lines 271–274 in the revised).

*Fig. 1 presents a comprehensive overview of the in-situ and lidar observation results over Beijing from 19:30 on November 2, 2023, to 09:30 on November 4, 2023 (CST). It integrates near-surface particulate matter concentrations with vertically resolved atmospheric profiles to illustrate the temporal evolution of a winter haze event. The figure is composed of six panels (a-f) that collectively depict the key parameters driving aerosol dynamics.*

4. Line 244 and throughout the manuscript: replace "figure" with "Fig.".

**Answer:** We appreciate the reviewers' careful review. We have systematically revised the entire manuscript by replacing all instances of the full spelling "figure" with the standard abbreviation "Fig.". This revision ensures consistency with academic writing conventions for figure citations throughout the paper.

5. Line 245: Abbreviation "RCS532" is explained only later in line 255.

**Answer:** The manuscript was modified accordingly (Line 278 in the revised).

6. Line 249: replace "typical" with "irregularly-shaped".

**Answer:** In the revised manuscript, we have streamlined and optimized the descriptions in the Results section. Consequently, the sentence originally at Line 249 in the revised has been rephrased and refined, making the requested replacement of the term "typical" with "irregularly-shaped" no longer applicable.

7. Line 325: "played crucial roles" to what- please rephrase or complete the sentence.

**Answer:** We have revised the sentence to explicitly state what these meteorological factors were crucial for, which is the "rapid clearance of pollutants." The intended meaning is now clearly articulated (Lines 316–320 in the revised).

*During the period of rapid pollutant clearance (before 2:00 CST on November 3rd), both the atmospheric boundary layer height and wind direction were crucial for the efficient removal of pollutants. The decrease in boundary layer height prevented the mixing of dust aerosols from above into the surface layer, and the strong northwesterly winds facilitated the rapid dispersion of anthropogenic pollutants.*

8. Line 331: legends states that PM observations start from 00:00 on Nov 2 and the lidar data from 19:30 and yet the plots are right above each other, giving the false impression that they refer to the same time period. The end time also differs. Please correct this taking into account the major comments above.

**Answer:** We have thoroughly revised the legend of Figure 1 to eliminate any possible misunderstanding regarding the time coverage of the dataset. The PM observation data and lidar data presented in Figure 1 belong to the same overall time frame, spanning from 19:30 CST on November 2, 2023, to 09:30 CST on November 4, 2023 (Line 309 in the revised).

9. Line 350: "based on the depolarization signal …" please mention again the method, i.e., POLIPHON.

**Answer:** We have revised the relevant text at Line 335 to explicitly reiterate the POLIPHON (Polarization Lidar Photometer Retrieval) method when describing the separation of dust and non-dust aerosols based on the depolarization signal (Line 335 in the revised).

*To gain an in-depth understanding of the transport behavior of aerosols from different sources, we have separated dust and non-dust aerosols following the POLIPHON method based on the depolarization signal (Fig. 2(b)).*

10. Lines 350-353: Dust is not predominantly located at higher altitudes, i.e., on Nov 2 and Nov 3 (around 12:00 CST) it reaches the "surface". Please rephrase the statement accordingly, including height boundaries (e.g., lower atmospheric layer).

**Answer:** We have revised the text in Lines 335-340 to provide a more precise description of the aerosol stratification, including specific height boundaries. This revision more accurately describes the vertical structure shown in Figure 2(b), acknowledging that while dust was mainly aloft, there were periods of significant mixing to the surface (Lines 335-340).

*It indicates that dust aerosols were primarily concentrated in the elevated layer (approximately 1-3 km), although they were occasionally mixed down to near the surface (e.g., on November 2 and around 12:00 CST on November 3). In contrast, non-dust aerosols were predominantly found in the lower atmospheric layer (below 1 km), which is mainly attributed to local anthropogenic emissions in the North China Plain. This classification enables us to investigate the transport fluxes of dust and non-dust aerosols separately.*

11. Figure 3 is very difficult to read. Please make the central map smaller and enlarge and enumerate all the plots around it. This will also improve the discussion in lines 354-361. The x-axis legend is missing.

**Answer:** We appreciate the reviewer's constructive comment on improving the readability of Figure 3. In response, we have revised the layout and design of Figure 3 comprehensively to enhance its clarity and information presentation.

[Figure]

**Figure 3:** The hourly air quality trends of major cities in the Beijing-Tianjin-Hebei region (Beijing, Tianjin, Zhangjiakou, Chengde) from November 2 to 4, 2023 are presented. The map (© Mapbox, available at https://www.mapbox.com/mapbox-studio) illustrates the geographical distribution of the cities, with surrounding subplots showing the daily variation patterns of PM$_{2.5}$ (black line), PM$_{10}$ (blue line) concentrations, and the PM$_{2.5}$/PM$_{10}$ ratio (pink dashed line) for each city. The concentration units are micrograms per cubic meter (µg/m³), and the time span is from 0:00 CST to 23:00 CST daily.

12. Line 367: remove the double dot.

**Answer:** The double dot was deleted.

13. Line 370: if possible, use either CST or UTC

**Answer:** We have standardized the time notation throughout the manuscript. Specifically, we have uniformly adopted China Standard Time (CST) for all time references related to the observation period and data analysis (Line 356 in the revised).

14. Line 421: "Compared to the study of near-surface aerosol fluxes" what do the authors refer to? If to another study it should be clearly stated and cited.

**Answer:** We have revised the relevant statement to explicitly specify the scope of the comparison and supplemented appropriate citations (Line 405 in the revised).

*Compared to the study of near-surface aerosol fluxes (e.g., Casquero-Vera et al., 2022; Pryor et al., 2008; Yuan et al., 2019), the vertical cross-sectional aerosol fluxes in this research are particularly useful for understanding the inter-regional transport of aerosols.*

15. Line 440-441: "indicating that… field" that is expected, please rephrase the statement

**Answer:** We have rephrased the original statement for greater accuracy and conciseness to avoid redundancy, while retaining the core scientific implication (Line 424-425 in the revised).

*The horizontal emission rate of non-dust aerosols (Fig. 6(c)) exhibited a trend highly synchronized with that of dust aerosols, underscoring their co-variation in response to the same wind-field forcing.*

16. Lines 474-477: To which figure do the authors refer to? The description provided does not match any of the subplots of Fig. 7. Please clarify.

**Answer:** We acknowledge that the original text contained an error in the figure citation and imprecise content description. In the revised manuscript, we have corrected the reference to Figure 7(b) (near-surface vertical column transport fluxes in the 100-300 m layer) and refined the corresponding analysis content for accuracy and clarity (Lines 460-468 in the revised).

*The comparison of near-surface vertical fluxes (100-300 m, Fig. 7 (b)) reveals distinct behaviors between dust and non-dust aerosols, indicative of their different sources. The vertical flux of dust exhibits strong intermittent deposition and occasional strong upward transport. In contrast, the vertical flux of non-dust aerosols remains consistently close to zero, with much smaller amplitudes of both positive and negative fluctuations throughout the event. This indicates that in the near-surface layer, the net vertical exchange of locally emitted non-dust aerosols is minimal, whereas dust undergoes significant episodic deposition and lifting, driven by synoptic-scale processes. This pattern supports the interpretation that dust is primarily subject to long-range transport and dynamic meteorological forcing, while non-dust aerosols are dominated by local, near-surface emissions with limited net vertical exchange on the event scale.*

17. Lines 478-480: Again, please mention the figure you refer to.

**Answer:** We have referenced the relevant figure (Line 469 in the revised).

18. Please consider making your research data available by depositing them in a FAIR-aligned repository (e.g., Zenodo).

**Answer:** We appreciate the reviewer's suggestion regarding data sharing. Due to CMA's restrictions, the research data presented in this study cannot be shared via a public FAIR-aligned repository for the time being. Nevertheless, for the sake of academic discussions and non-commercial research use, interested researchers are welcome to contact the corresponding author (longlong.wang@zjnu.edu.cn) to request access to the relevant data.

19. I have noticed several issues with the literature section, please make sure all your entries are listed correctly.

**Answer:** We thank the reviewer for their careful evaluation of the literature section. In response, we have added the relevant literature and carried out a full-scale review and revision of all reference entries in the manuscript to guarantee their accuracy and consistency.

**References**

Huang, R.-J., Zhang, Y., Bozzetti, C., Ho, K.-F., Cao, J.-J., Han, Y., Daellenbach, K. R., Slowik, J. G., Platt, S. M., Canonaco, F., Zotter, P., Wolf, R., Pieber, S. M., Bruns, E. A., Crippa, M., Ciarelli, G., Piazzalunga, A., Schwikowski, M., Abbaszade, G., Schnelle-Kreis, J., ... Pre´voˆt, A. S. H. (2014). High secondary aerosol contribution to particulate pollution during haze events in China. Nature, 514(7523), 218–222. https://doi.org/10.1038/nature13774

Liu, Q., Baumgartner, J., Zhang, Y., & Schauer, J. J. (2016). Source apportionment of Beijing air pollution during a severe winter haze event and associated pro-inflammatory responses in lung epithelial cells. Atmospheric Environment, 126, 28–35. https://doi.org/10.1016/j.atmosenv.2015.11.031

Wandinger, U., Linné, H., Bösenberg, J., Eromskis, E., Althausen, D., & Müller, D. (2004, June). Turbulent aerosol fluxes determined from combined observations with Doppler wind and Raman aerosol Lidar. In 22nd Internation Laser Radar Conference (ILRC 2004) (Vol. 561, p. 743).

Zhang, R., Jing, J., Tao, J., Hsu, S.-C., Wang, G., Cao, J., Lee, C. S. L., Zhu, L., Chen, Z., Zhao, Y., & Shen, Z. (2013). Chemical characterization and source apportionment of $PM_{2.5}$ in Beijing: Seasonal perspective. Atmospheric Chemistry and Physics, 13(11), 7053–7074. https://doi.org/10.5194/acp-13-7053-2013